# Field-free spin-orbit torque switching via out-of-plane spin-polarization induced by an antiferromagnetic insulator/heavy metal interface

Mengxi Wang[1,6], Jun Zhou[2,6], Xiaoguang Xu[1] ✉, Tanzhao Zhang[1], Zhiqiang Zhu[1], Zhixian Guo[1], Yibo Deng[1], Ming Yang [3] ✉, Kangkang Meng[1], Bin He[4], Jialiang Li[4], Guoqiang Yu [4], Tao Zhu[4], Ang Li [5], Xiaodong Han [5] & Yong Jiang [1] ✉

Manipulating spin polarization orientation is challenging but crucial for field-free spintronic devices. Although such manipulation has been demonstrated in a limited number of antiferromagnetic metal-based systems, the inevitable shunting effects from the metallic layer can reduce the overall device efficiency. In this study, we propose an antiferromagnetic insulator-based heterostructure NiO/Ta/Pt/Co/Pt for such spin polarization control without any shunting effect in the antiferromagnetic layer. We show that zero-field magnetization switching can be realized and is related to the out-of-plane component of spin polarization modulated by the NiO/Pt interface. The zero-field magnetization switching ratio can be effectively tuned by the substrates, in which the easy axis of NiO can be manipulated by the tensile or compressive strain from the substrates. Our work demonstrates that the insulating antiferromagnet based heterostructure is a promising platform to enhance the spin-orbital torque efficiency and achieve field-free magnetization switching, thus opening an avenue towards energy-efficient spintronic devices.

Magnetization switching driven by spin-orbit torque (SOT) is a promising technology for low-power, high-speed logic and storage devices[1–6]. However, traditional SOT structures such as Pt/Co/Al$_2$O$_3$ and Ta/CoFeB/MgO require an auxiliary magnetic field to switch the perpendicularly magnetized layer, which significantly constrains device integration and scalability[7]. As a result, there is a need to develop a method for achieving zero-field magnetization switching (ZFS)[8] and various attempts have been made to achieve this goal. One method is to introduce an effective field along the in-plane direction using the exchange bias effect from an antiferromagnetic layer[9–11]. The effective field functions as the auxiliary magnetic field to tilt the perpendicular magnetization state and achieve ZFS. Another method is to utilize two competing spin currents generated in two heavy metal (HM) layers on the bottom of the magnetic layer. The opposite spin Hall angles ($\theta_{SH}$) of the HM layers create two spin currents with opposite polarization directions, facilitating ZFS[12]. In addition, ZFS can be realized in wedge architectures by introducing gradient spin current[13–15].

[1]School of Materials Science and Engineering, University of Science and Technology Beijing, 100083 Beijing, China. [2]Institute of Materials Research and Engineering (IMRE), Agency for Science, Technology and Research (A*STAR), 2 Fusionopolis Way, Innovis #08-03, Singapore 138634, Republic of Singapore. [3]Department of Applied Physics, The Hong Kong Polytechnic University, Hong Kong SAR, China. [4]Beijing National Laboratory for Condensed Matter Physics, Institute of Physics, Chinese Academy of Sciences, 100190 Beijing, China. [5]Faculty of Materials and Manufacturing, Beijing Key Lab of Microstructure and Properties of Advanced Materials, Beijing University of Technology, 100124 Beijing, China. [6]These authors contributed equally: Mengxi Wang, Jun Zhou. ✉e-mail: xgxu@ustb.edu.cn; kevin.m.yang@polyu.edu.hk; yjiang@ustb.edu.cn

Typically, spin-current and spin-polarization ($\sigma_s$) resulting from a charge current are mutually orthogonal. Thus, a charge current along the *x*-axis with a spin-current flowing along the *z*-axis accompanies with a spin-polarization along the y-axis. However, a $\sigma_s$ deviating from the *y* axis is required to achieve ZFS. Recently, an approach was proposed to generate a spin polarization with a component perpendicular to the plane defined by the directions of the charge and spin current[16–20], referred to as "out-of-plane". This kind of spin current has been demonstrated in ferromagnetic (FM) trilayers[16,17] and single-crystal WTe$_2$[18]. Specifically, the $\sigma_s$ with an out-of-plane component can exist in the non-collinear antiferromagnets Mn$_3$GaN[19] and Mn$_3$Sn[20] as well as the collinear antiferromagnet Mn$_2$Au[21]. However, due to the metallic nature of these antiferromagnetic layers, the current will flow through both the antiferromagnetic and ferromagnetic layers, leading to a current shunting effect that decreases overall device efficiency. Besides, the relatively low Néel temperatures of the above-mentioned AFM metals could be another limitation for the practical applications, as AFM ordering becomes unstable near the Néel temperature, particularly in the presence of a current. Therefore, an efficient approach to generating spin polarization with an out-of-plane component and a high Néel temperature is required for next-generation high-performance spintronic applications.

In this work, we propose a stacking structure based on insulating antiferromagnet/HM heterostructures to achieve a spin polarization with an out-of-plane component. Using NiO/Ta/Pt/Co/Pt as an example, we demonstrate that an out-of-plane polarization component $\sigma_s$ ($\sigma_{sz}$) can be introduced by the NiO/HM interface, enabling ZFS. Moreover, the insulating nature of NiO prevents the current shunting effect. Our experiments and first-principles calculations confirm that the ZFS ratio can be effectively tuned by different substrates. Notably, the Néel temperature of the antiferromagnetic insulator NiO is high, reaching up to 520 K[22], which enables operation of this spintronic device at higher temperature.

## Results

We utilized pulsed layer deposition (PLD) to epitaxially grow NiO(001) layers on both MgO(001) and SrTiO$_3$(001) substrates to investigate the performance of NiO-based spin-orbit torque (SOT) devices. Figure 1a–d shows X-ray diffraction (XRD) and cross-sectional transmission electron microscopy (TEM) images of MgO(001)/NiO(20)/Ta(0.2)/Pt(4) (the numbers in brackets indicate the thickness in nanometers), revealing the high crystalline quality of the NiO layers. Additionally, the insulating nature of the NiO layer is demonstrated by the I-V curve (see Supplementary Fig. 1), which ensures avoidance of the current shunting effect in the NiO layer, in contrast with that in antiferromagnetic metal layers as proposed in previous studies. Next, we deposited multilayers of Ta(0.2)/Pt(4)/Co(1)/Pt(1) on MgO (STO)/NiO(20) using magnetron sputtering. Control groups were also fabricated without the NiO layer, consisting of MgO(001)/Ta(0.2)/Pt(4)/Co(1)/Pt(1) and STO(001)/Ta(0.2)/Pt(4)/Co(1)/Pt(1) structures. These samples will be referred to as MgO(001)/NiO(20), STO(001)/NiO(20), MgO(001), and STO(001) samples in the subsequent discussions. An ultrathin Ta(0.2) layer was inserted to stabilize the perpendicular magnetic anisotropy (PMA) of the Pt/Co/Pt multilayers.

The samples were patterned into Hall bars with a size of 20 μm × 10 μm by electron beam lithography (EBL) and etching process, as illustrated in Fig. 2a. The perpendicular direction to the plane is defined as *z*, current is along the *x* direction, and lateral Hall voltage is measured along the *y* direction. As shown in Fig. 2b–e, the anomalous Hall effect (AHE) curves show the change of the anomalous Hall resistance ($R_H$) with respect to the magnetic field, which is the characteristic feature of multilayers with PMA. To investigate the role of NiO layers in magnetization switching, we compared SOT-based switching in MgO(001) and MgO(001)/NiO(20) samples under various magnetic fields. Figure 3a, b reveals that the MgO(001)/NiO(20) sample can achieve current-induced ZFS, unlike the MgO(001) sample. By comparing Fig. 3c with d, we observe that the ZFS ratio is around 0%

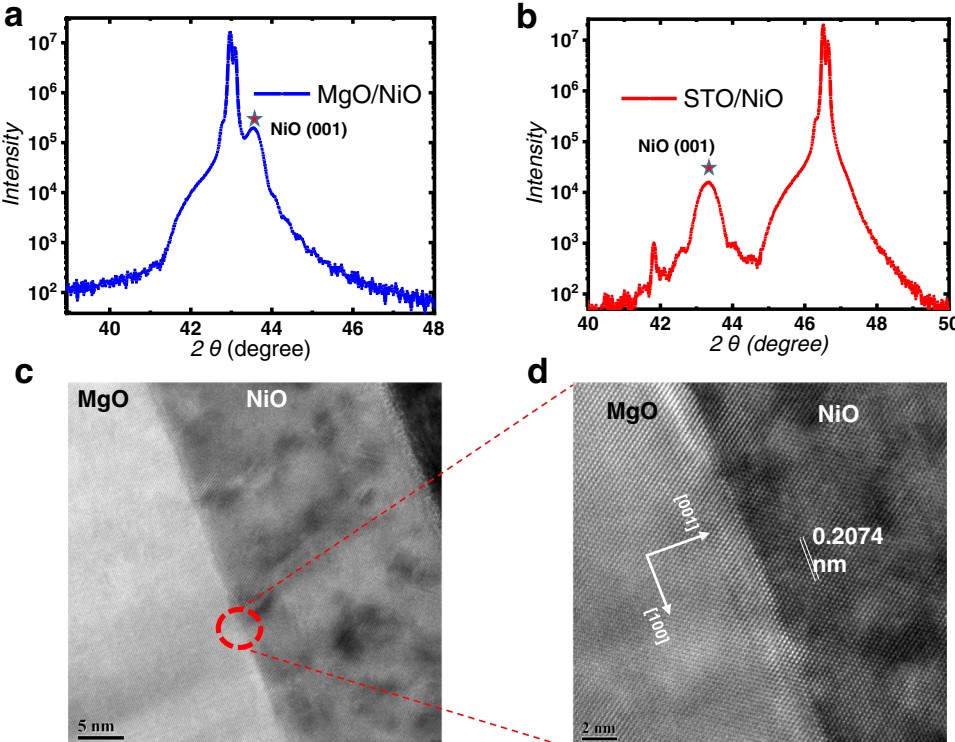

**Fig. 1 | Structure characteristics of NiO on different substrates. a, b** The X-ray diffraction patterns of the NiO(20 nm) film on the MgO(001) and STO(001) substrates, respectively. The pentacle marks out the peak of NiO(001). **c** Cross-sectional TEM image of the structure of MgO(001)/NiO(20)/Pt. **d** HRTEM image of the MgO(001)/NiO(20) interface.

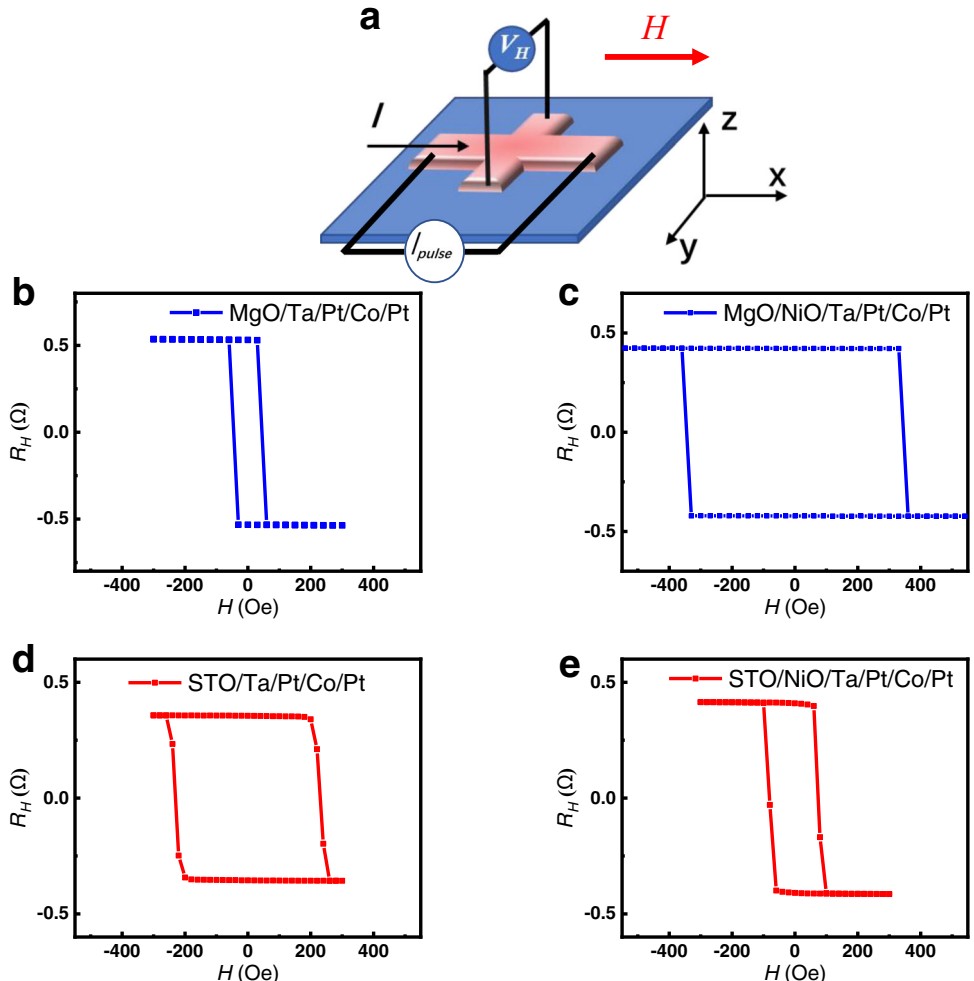

**Fig. 2 | Anomalous Hal voltage hysteresis of different structures. a** The scheme of Hall bar and the relationship of the direction of current flow and Hall voltage. $I_{pulse}$ represents the pulse current, $V_H$ represents the hall voltage, $H$ represents the magnetic filed and the red arrow indicates its direction. **b–e** The AHE loop of the

Hall bar with the stacking structure of MgO(001)/Ta(0.2)/Pt(4)/Co(1)/Pt(1), MgO(001)/NiO(20)/Ta(0.2)/Pt(4)/Co(1)/Pt(1), STO(001)/Ta(0.2)/Pt(4)/Co(1)/Pt(1) and STO(001)/NiO(20)/Ta(0.2)/Pt(4)/Co(1)/Pt(1), respectively.

for MgO(001) and nearly 50% for MgO(001)/NiO(20). Additionally, we performed MOKE measurements on the MgO(001)/NiO(20) sample to demonstrate SOT-based ZFS. As shown in Fig. 3e–h, the impulse current's direction caused domain inversion without an external magnetic field, providing direct evidence for SOT-driven ZFS. These results provide compelling evidence for the crucial role of the insulating NiO layer in achieving ZFS.

Next, we explore the mechanism that is responsible for the magnetization switching. It is known that the exchange bias field can serve as an auxiliary magnetic field to achieve ZFS[9–11], thus the exchange coupling effect between NiO and Co might contribute to the ZFS. However, in our samples a thick Pt layer (4 nm) was inserted between the Co and NiO layers to minimize the exchange bias effect. And the hysteresis loops for the in-plane and out-of-plane directions of the samples (See Supplementary Fig. 2) also clearly show the absence of exchange bias in the MgO(001)/NiO(20) sample. Therefore, we can rule out the contribution of the exchange bias between the NiO and Co layers to the ZFS in this device. Furthermore, comparing with previous reports about the Néel vector switching of NiO[23–26], we only applied a pulse current along one direction, thus the variation of the resistance caused by electron migration and Joule heating should not have a significant contribution to the AHE and $R_H$-$I$ loops. The loop-like signal is not observed in the samples without Co layers (See Supplementary Fig. 3). Therefore, it can be concluded that

the loop-like curves are only a response to the magnetization switching of the magnetic Co layer. On the other hand, the predicted critical current threshold for NiO Néel order switching is up to $5.8 \times 10^{12}$ A m$^{-2}$[26], which is nearly one order of magnitude larger than that in this work (a maximum current of $6 \times 10^{11}$ A m$^{-2}$). Previous experiments have demonstrated the switching of Néel order using a lower threshold current, however, this was accomplished using a longer pulse width of up to 1 ms[24,26]. In contrast, our study utilized a low-current pulse with a short pulse width of 16 μs, which significantly reduces the likelihood of Néel vector switching during Co magnetization switching. Therefore, we do not anticipate the reorientation of the Néel vector in our measurements.

Another possible mechanism for the ZFS is that $\sigma_s$ has an out-of-plane component $\sigma_{sz}$, which has been proposed in several recent reports[16–21]. To unravel this effect, we study the AHE loops of the MgO(001)/NiO(20)/Ta(0.2)/Pt(4)/Co(1)/Pt(1) sample with the variation of the direction and magnitude of the applied current. As shown in Fig. 4a–d, the center of each AHE loop begins to exhibit a relative offset when the current is as large as ±4 mA, and the absolute value of the bias field further increases with the current. The center of the magnetization switching loops, defined as $H_O(I^\pm)$, can be calculated by $H_O(I^\pm) = [|H_c^+(I^\pm)| - |H_c^-(I^\pm)|]/2$, and the bias field $\Delta H(I)$ is obtained by $|\Delta H(I)| = |H_O(I^+) - H_O(I^-)|$. The absolute values of $|\Delta H(I)|$ with respect to the magnitude of the current are summarized in Fig. 4d. A clear critical

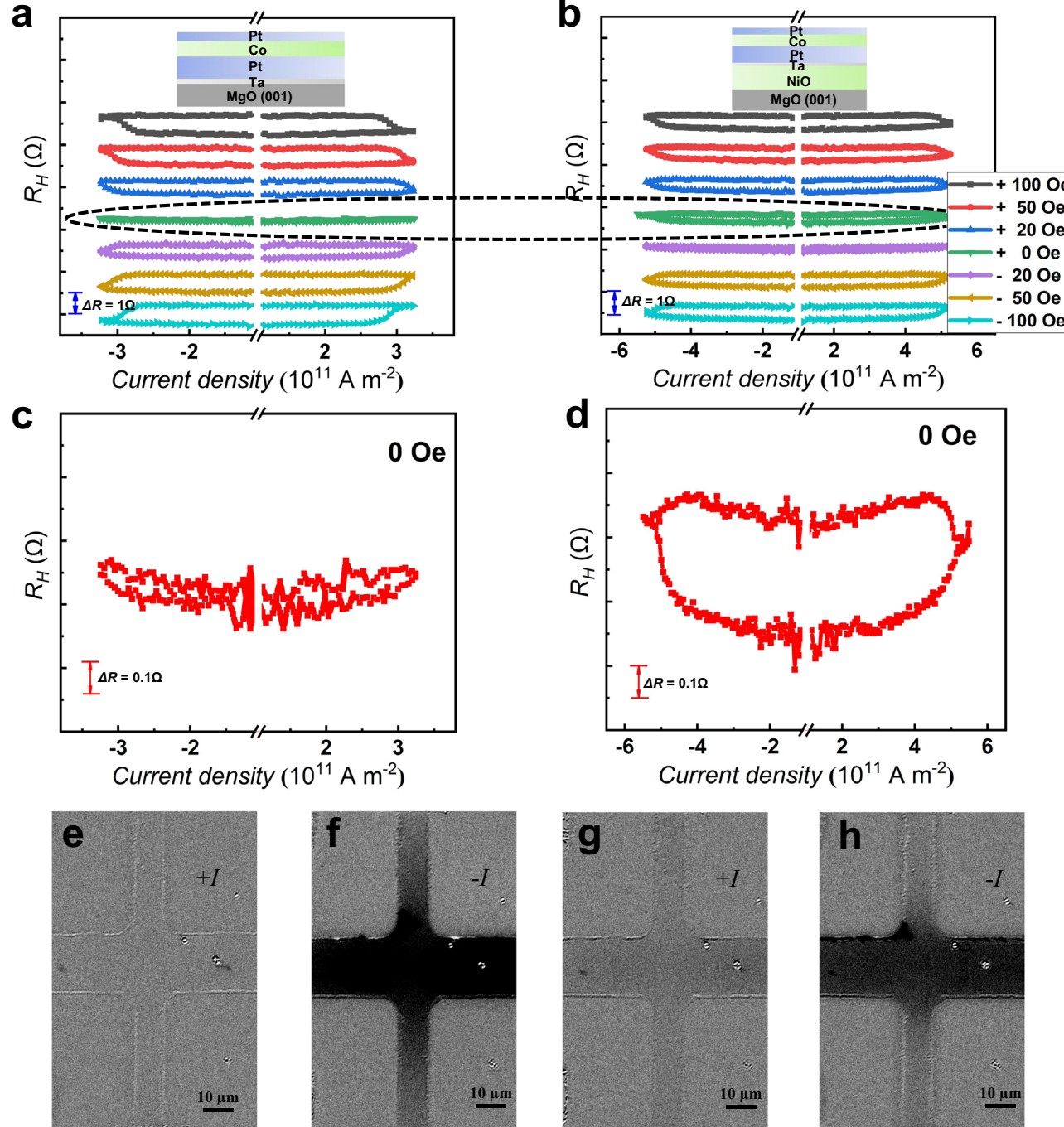

**Fig. 3 | Current-driven SOT switching. a, b** The SOT-based magnetization switching curves of the MgO(001) and MgO(001)/NiO(20) samples under a series of auxiliary magnetic fields, respectively. **c, d** show the ZFS curves of the MgO(001) and MgO(001)/NiO(20) samples, respectively. **e–h** MOKE images of the magnetization switching in MgO(001)/NiO(20) sample post applying different directions of impulse current without the assistance of magnetic field. +*I* and −*I* represent the different directions of the applied current.

state transition can be observed at around 3 mA. When the current is less than 3 mA (see more details of the $R_H$-$H$ curves for ±1 and ±2 mA in Supplementary Fig. 4), the center of the AHE loop does not have relative deviation for the positive and negative current.

The phenomenon can be explained as follows. In the perpendicularly magnetized structure, a spin current with $\sigma_{sz}$ will generate an anti-damping like torque on the magnetic ordering[16]. When the current is large enough, the anti-damping like torque exceeds the intrinsic damping like torque, leading to the threshold phenomenon. The center of the AHE loop will shift with the change of the current

direction and result in the offset of the bias field, as shown in Fig. 4d. This critical phenomenon combined with the current-induced ZFS confirms the existence of $\sigma_{sz}$ in the heterostructure. $\sigma_{sz}$ breaks the symmetry of the magnetic ordering in the ferromagnetic layer with PMA, which enables the magnetization switching occur preferably in one direction depending on the direction of the current, resulting in the asymmetric $R_H$-$H$ curves. In contrast, in the traditional structure without $\sigma_{sz}$, the center of the AHE loop should not shift with the current direction without the assistance of a magnetic field along the $x$ direction. After applying an external magnetic field along the $x$

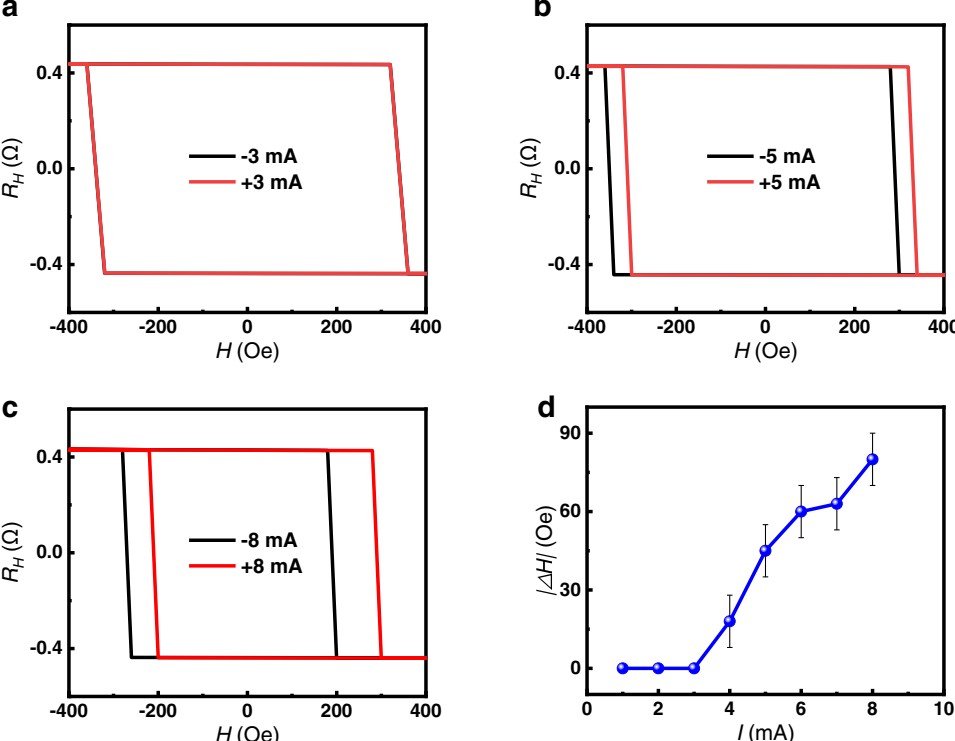

**Fig. 4 | Out-of-plane polarized spin current generated by antiferromagnetic interface. a–c** The AHE loops with the applied current of ±3 mA, ±5 mA and ±8 mA on the MgO(001)/NiO(20)/Ta(0.2)/Pt(4)/Co(1)/Pt(4) sample, respectively. **d** The shift of $|\Delta H(I)|$ with respect to the magnitude of the applied current. The error bar comes from the increment of magnetic field during the measurement.

direction, a $|\Delta H(I)|$ proportional to the current will appear for any given magnitude, which is different from the results we observed[16].

To further demonstrate the contribution of $\boldsymbol{\sigma_{sz}}$ to the ZFS, we fabricated another two samples with the same thickness (4 nm) of the top and bottom Pt layers on MgO(001) and MgO(001)/NiO(20), respectively. As shown in Supplementary Fig. 5a, d, both the AHE results display PMA. However, when the thickness of the top and bottom Pt layers is the same, the spin accumulation generated at the upper and lower interfaces between the Pt and Co layers will have the same magnitude but in opposite directions, which will cancel each other out. Therefore, as shown in Supplementary Fig. 5b, c, the magnetization switching cannot be achieved even under a large auxiliary field of 800 Oe. In contrast, when the NiO layer is inserted between the MgO substrate and Ta layer, the $\boldsymbol{\sigma_{sz}}$ of the spin current at the interface deviates from the in-plane direction and has an out-of-plane component. Therefore, the SOT-driven magnetization switching can be observed at ±500 Oe, as shown in Fig. 5e, f.

For comparison, we also grew epitaxial NiO(001) with different thicknesses on STO substrate and fabricated the Hall bar structures. Figure 5 shows the $R_H$-$I$ measurement results of the MgO(001)/NiO(5/20 nm)/Ta(0.4)/Pt(4)/Co(1)/Pt(1) and STO(001)/NiO(5/20 nm)/Ta(0.4)/Pt(4)/Co(1)/Pt(1) heterostructures. For NiO layer with a thickness of 20 nm, the sample with STO substrate can also achieve the SOT induced ZFS. However, comparing with the samples on the MgO substrate, the ratio of ZFS decreases from 50% to 40%, as shown in Fig. 5a, b. For the cases with 5 nm NiO layer, the ZFS ratio of the MgO substrate sample is about 52% (see Fig. 5c), which is comparable with that of the thick NiO sample. While for the sample grown on the STO substrate, the ratio of ZFS dramatically decreases to about 21% (see Fig. 5d), which is much less than that of the STO/NiO(20 nm) sample.

To clarify the interfacial magnetic order of stacking structures, we have characterized the samples of MgO(001)/NiO(5)/Ta(0.2)/Pt(4)/Co(1)/Pt(1) and STO(001)/NiO(5)/Ta(0.2)/Pt(4)/Co(1)/Pt(1) by using

polarized neutron reflectivity (PNR) measurements. Figure 6a, b show the nuclear scattering length density (nSLD) and magnetization scattering length density (mSLD), which represent the chemical and magnetization profiles of the sample, respectively. Figures 6c, d show the splitting between the spin-up and spin-down neutrons corresponding to the magnetic contribution from the in-plane direction of Co layer, as the PNR is only sensitive to in-plane magnetization. However, due to the large external field, the perpendicular Co magnetic moments can be tilted, resulting in the splitting[27,28]. It is obvious that the magnetization only comes from the Co layer and the NiO layer is antiferromagnetic with zero magnetization. According to the PNR results, the NiO in our sample can be confirmed to be antiferromagnetic even at the thickness of about 5 nm. Moreover, we can obtain the compensated antiferromagnetic interface in the MgO(001)/NiO(5) sample by PNR measurement. As illustrated in Fig. 6a, c, there is no obvious net magnetic moments observed at the NiO/HM interface, which indicates that the uncompensated magnetic moments are negligible. While in the PNR measurement of STO(001)/NiO(5)/Ta(0.2)/Pt(4)/Co(1)/Pt(1) sample, as presented in Fig. 6b, d, the negative net spins can be obtained at the NiO/HM interface on STO(001)/NiO(5) substrate. It indicates that the interface of NiO/HM is uncompensated in the STO(001)/NiO(5) samples.

These results can be explained as follows. NiO is a collinear antiferromagnetic insulator. Due to the work function difference between the NiO and Pt, there is an internal electric field pointing from NiO to Pt[29]. When a current (denoted as $J_e$) flows through the HM layer, it can be converted into a spin current (denoted as $J_s$) via the spin Hall effect. The initial polarization direction of the spin current is perpendicular to the current $J_e$ and in the in-plane direction. At the NiO/HM interface, the internal electric field interacts with $J_s$, leading to spin flip, rotation, and precession, resulting in the reorientation of the spin polarization with an out-of-plane component. The out-of-plane polarized spin current will be reflected to the Co layer and drive the ZFS. In

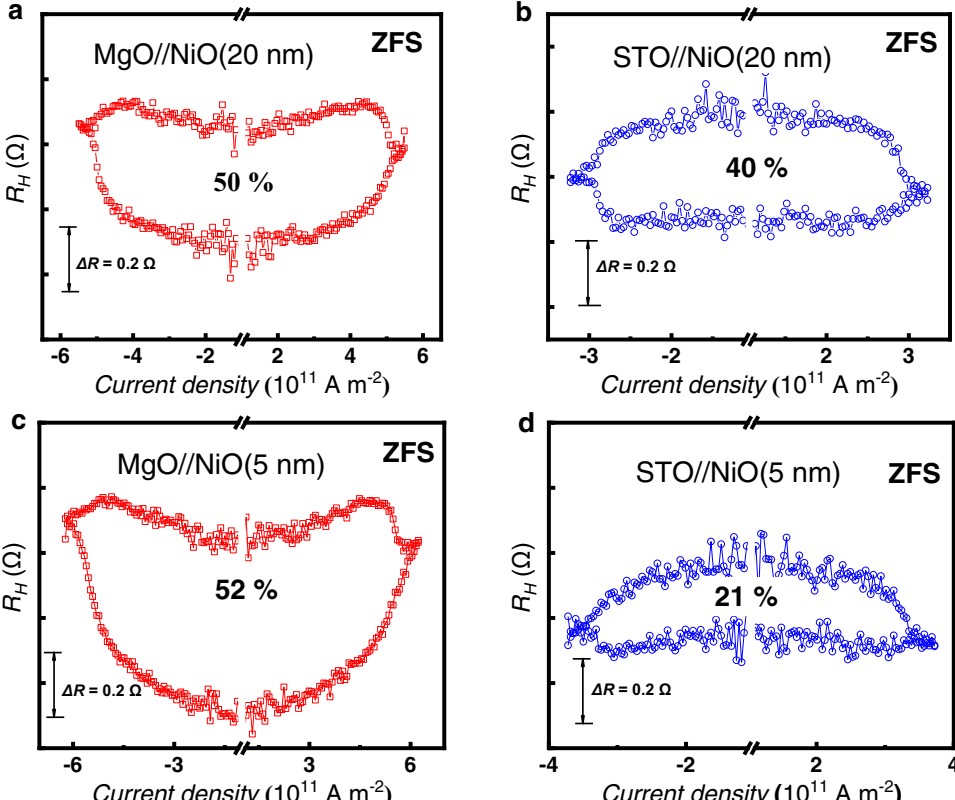

**Fig. 5 | The influence of lattice strain on SOT switching ratio. a–d** show the SOT induced ZFS of perpendicular magnetization in the samples of MgO(001)/NiO(20)/Ta(0.2)/Pt(4)/C (1)/Pt(1), STO(001)/NiO(20)/Ta(0.2)/Pt(4)/Co(1)/Pt(1), MgO(001)/ NiO(5)/Ta(0.2)/Pt(4)/Co(1)/Pt(1) and STO(001)/NiO(5)/Ta(0.2)/Pt(4)/Co(1)/Pt(1) respectively. ZFS represents the magnetization switching occurred without the assistance of magnetic field.

the MgO(001) sample, the process of generation and transportation of z-polarized spin current are illustrated in Fig. 6e. In this case, the interface of NiO/HM contains both spin up and spin down Ni moments, so the magnetic moments cancel each other out and cannot contribute net spins. Consequently, the z-polarized spin current induced by the internal electric field in the MgO(001) sample reflects back at the compensated interface without absorption. Furthermore, the thickness of the bottom Pt layer (4 nm) is less than the typical spin diffusion length of Pt (4−10 nm)[30], enabling the transport of the $J_s$ with $\sigma_{sz}$ to Co layer to drive the ZFS. In the stacking structure, the thickness of the HM Pt layer between NiO and Co is critical for realizing the $\sigma_{sz}$-driven ZFS. A Pt thickness of 4 nm is thick enough to minimize exchange bias effects between NiO and Co but thin enough to maintain spin current transport.

Third, we reveal the substrate strain effect on the Néel order of NiO. The substrate MgO (with a lattice parameter of 4.21 Å) enforces a slight tensile strain on NiO (with a lattice parameter of 4.17 Å), while the substrate STO (with a lattice parameter of 3.91 Å) imposes a larger compressive strain on NiO[31,32]. In the STO(001)/NiO(5) sample, the Néel vector of NiO tilts from out-of-plane to in-plane under the compressive strain and the NiO/HM interface tends to turn into uncompensated[31]. The z-polarized spin current will be absorbed partially at the uncompensated interface, which means less z-polarized spin current will act on the Co layer (shown in Fig. 6f). Therefore, the different interactions of the antiferromagnetic interfaces are responsible for the changed ZFS ratio in MgO(001) sample and STO(001) sample. In addition, when the NiO film is thinner, the influence of the lattice strain from STO will have a more significant impact on the Néel vector direction of NiO(5 mn). In contrast, when a thicker NiO(20 nm) layer is used, the stress from the substrate will be released. In such case, the Néel vector tends to lie in the (111) plane and the resulting compensated interface will

lead to a larger ZFS ratio (see in Fig. 5b, d). Comparing to the STO substrate, the crystal lattice of NiO matches better with the MgO substrate (around 1% mismatch). As a result, the NiO layer suffers smaller lattice strain and can retain the Néel vector in the (111) plane. Therefore, the ZFS ratio of the sample on the MgO substrate is insensitive to the thickness of NiO. Furthermore, to compare the quality of NiO layer growth on different substrates, we have studied the surface of NiO and NiO/Pt films by atomic force microscopy (AFM). The AFM images of the as-deposited NiO and NiO/Pt films on MgO and STO substrates are presented in Supplementary Fig. 6. It can be seen that the films are smooth with a $R_q$ value lower than 0.2 nm after NiO layer growth and the roughness also remains comparable after the metal layers deposition. Meanwhile, the high crystal quality of the NiO layer and the NiO/Pt interface on MgO and STO substrates are confirmed by HRTEM, as shown in Supplementary Fig. 7.

Based on the above analysis, it can be concluded that the ZFS in our samples is related to the internal electric field and interfacial states of NiO/HM. To further verify this model, we also fabricated another sample with the stacking structure of MgO(111)/NiO(5)/Ta(0.2)/Pt (4)/ Co(1)/Pt(1). In this sample, the crystal orientation of epitaxially grown NiO is [111]. Therefore, the Néel vector of NiO is along the in-plane direction since the easy plane is along the (111). In this case, the SOT induced ZFS ratio (see in Supplementary Fig. 8) is comparable with that of the STO(001) sample. The switching ratio of ZFS is calculated to be about 18% on the MgO(111) substrates. Considering that the Néel vectors in MgO(111)/NiO(5) and STO(001)/NiO(5) samples are along the in-plane direction, the uncompensated interface will absorb the z-polarized spin current partially, similar to the STO(001) case.

Finally, in order to support our proposed mechanism of the NiO/HM interface-induced ZFS, we performed first-principles calculations for the NiO bulk with different in-plane strains. The schematic

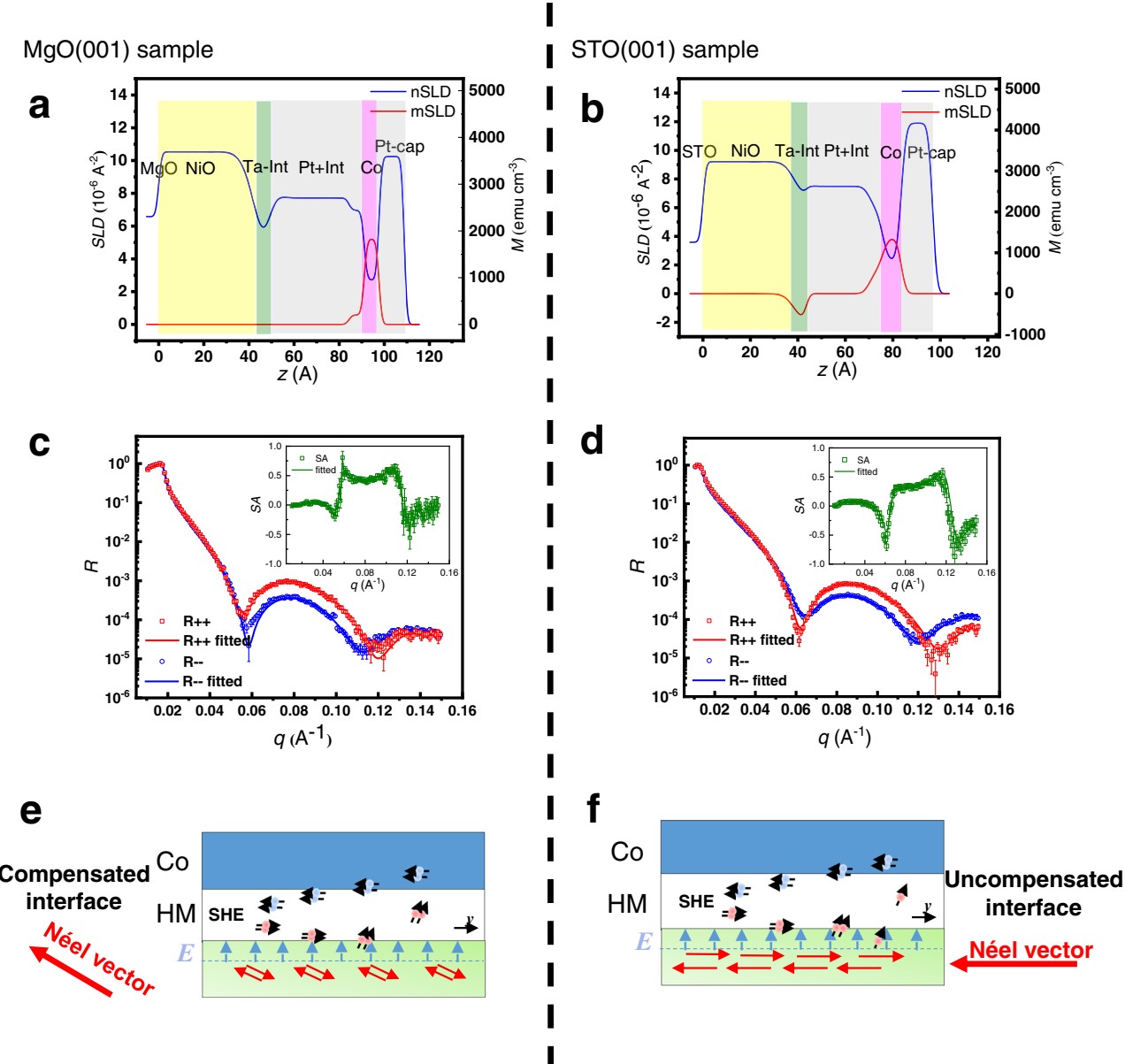

**Fig. 6 | Characteristics of NiO/HM interfaces. a, b** Nuclear scattering length density (nSLD) and magnetization profiles (mSLD) of the MgO(001)/NiO(5)/Ta(0.2)/Pt(4)/Co(1)/Pt(1) and STO(001)/NiO(5)/Ta(0.2)/Pt(4)/Co(1)/Pt(1) samples respectively; **c, d** The polarized neutron reflectivity ($R^{++}$ and $R^{--}$) of the MgO(001)/NiO(5)/Ta(0.2)/Pt and STO(001)/NiO(5)/Ta(0.2)/Pt(4)/Co(1)/Pt(1) samples as a function of wave factor (q) respectively; **e, f** show the generation and transportation of $z$-polarized spin current at the compensated and uncompensated interfaces. The red arrows represent the directions of Néel vectors, $E$ represents the internal electric field and the blue arrows indicate its direction. The blue balls and red balls represent the different directions of spin polarization. The error bars come from the fitting process in Fig. 6c, d.

diagrams of atomic structure and the magnetic moments of NiO are shown in Fig. 7a. Considering the non-polar and insulating nature of NiO, STO and MgO, the interface effects of MgO/NiO and STO/NiO are expected to be dominated by strain effects. The strain dependent magnetic easy axis can be identified by rotating the spins of all the Ni atoms simultaneously within the $(1\bar{1}0)$ plane and picking the spin alignment with the energetic ground state for each strain. The direction of the magnetic easy axis is indicated by its angle with the [110] orientation ($\theta$) as shown in the inset of Fig. 7b. The specific magnetization directions examined in this work and their magnetic anisotropy energies (MAE) are summarized in Supplementary Fig. 9. As shown in Fig. 7b, the magnetic easy axis of pristine NiO is along [112] orientation with a $\theta$ value of 54.73°, in line with the experimental observations[31]. Interestingly, a compressive strain tilts the magnetic easy axis to the [110] orientation, while a tensile strain enhances the out-of-plane

strength gradually and the magnetic easy axis even completely aligns along the [001] direction at 2% tensile strain. This is in line with the larger SOT ratio for NiO grown on MgO than that on STO.

The projected density of states (PDOS) for orbitals with out-of-plane component are shown in Fig. 7c. A strong hybridization between the Ni-$d_{xz}/d_{yz}/d_{z^2}$ and O-$p_z$ orbitals is evidenced by their similar profiles. Besides, the tensile (compressive) strain pushes these orbitals to higher (lower) energy level. According to the second-order perturbation theory, the perpendicular MAE is proportional to the energy of the occupied orbitals with out-of-plan component[33,34]. Thus, the change of magnetic easy axis can be understood by the shift of the Ni-$d_{xz}/d_{yz}/d_{z^2}$ and O-$p_z$ orbitals due to the substrate induced strains. As for the Hall bar on MgO substrate, the NiO is close to the pristine state, and the Néel vector lies in the [112] direction. Therefore, when the current flows through the Hall

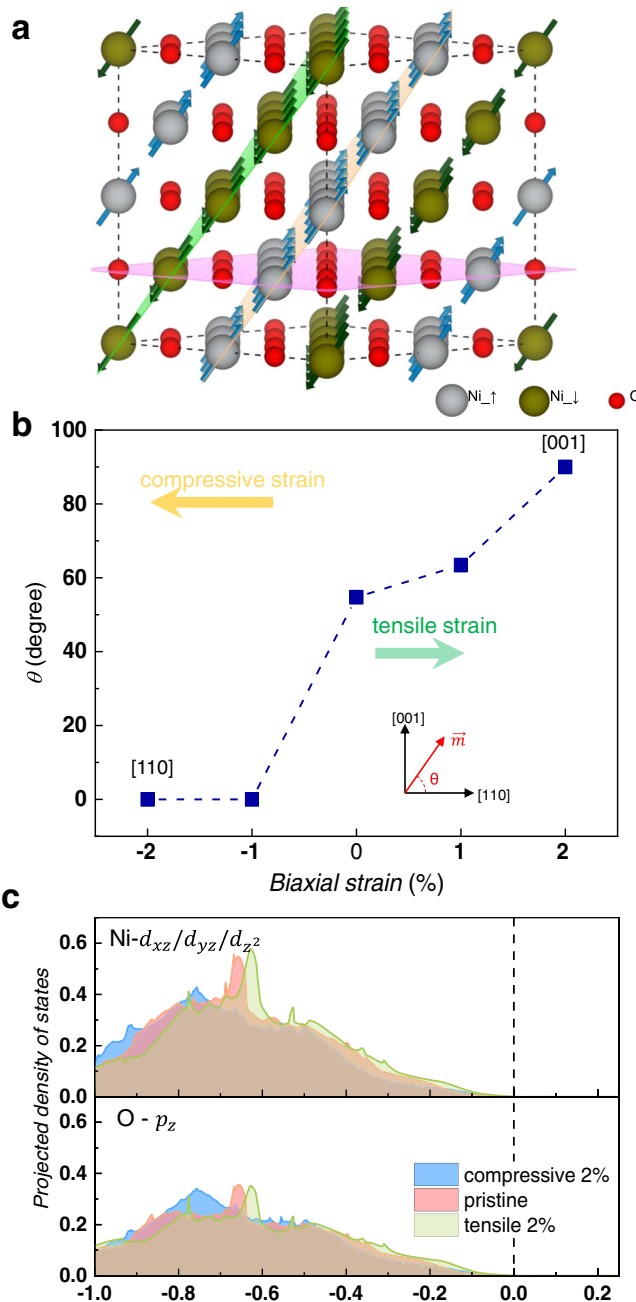

**Fig. 7 | First-principles calculations for the NiO under strains. a** Structural guidance of bulk NiO. The gray, dark green and red balls represent the spin-up and spin-down Ni as well as O atoms, respectively. **b** Biaxial strain dependent magnetic easy axis of NiO. The direction of magnetization ($\vec{m}$) was rotated within the ($1\bar{1}0$) plane by an angle of θ with the [110] direction as reference. **c** Biaxial strain dependent PDOS of Ni-$d_{xz}/d_{yz}/d_{z^2}$ and O-$p_z$ orbitals. It should be noted that the spin-down PDOSs are not shown for clarity as they identical to the spin-up components due to the antiferromagnetic nature of NiO.

**Table 1 | The comparison between our NiO based ZFS devices and previous AFM metals based ZFS devices**

| Items | This work | Previous works |
|---|---|---|
| Structure of devices | Antiferromagnetic insulator NiO-based device | Other AFM metallic structures[10,11,19,20] |
| ZERO-FIELD SWITCHING mechanism | Spin polarication with out-of-plane component | a) Exchange bias[10,11] |
| | | b) Competing spin current[12] |
| | | c) Spin polarization with out-of-plane component[16–20] |
| Critical switching current | 3 ~ 6 × 10¹¹ A m⁻² | a) 8 ~ 9 × 10¹¹ A m⁻² [10] |
| | | b) 3 × 10¹¹ A m⁻² [12] |
| | | c) 6 × 10¹¹ A m⁻² [20] |
| Néel temperature | 520 K[22] | 345 K (Mn₃GaN)[19], 475 K (Mn₃SnN)[20] |
| Shunting | No | Yes |
| Néel vector regulation | Yes | Yes |
| z-direction spin-polarization regulation | Yes (Lattice strain) | No |

higher Néel temperature of NiO, which enables the device to work at higher temperature, compared with those AFM metal based SOT devices. Besides, our work successfully demonstrates the tuning of the spin-polarization direction in the ZFS SOT devices.

## Discussion

In this study, we propose an antiferromagnetic insulator-based heterostructure to realize SOT induced perpendicular magnetization switching without the assistance of a magnetic field. The antiferromagnetic insulator used in the structures prevents the current shunting and induces the out-of-plane component of spin polarization due to the internal electric field at the interface of NiO/HM. Moreover, we unravel the effect of the substrate induced lattice strain on the Néel order of NiO layer and NiO/HM interface states, resulting in substantial tuning of ZFS. Our work demonstrates the field-free SOT devices based on antiferromagnetic NiO, which can increase the device efficiency, enhance the device stability to higher working temperature, and extend the limited choices of antiferromagnetic metals to a broad range of antiferromagnetic insulators, paving a promising avenue towards high-performance and low energy consumption memory.

## Methods

### Sample preparation

The NiO was deposited on MgO and STO substrates by pulsed laser deposition (PLD) at the temperature of 550 °C and the O₂ pressure of 5 Pa. The metal layers Ta, Pt and Co were deposited using magnetron sputtering under the Ar pressure of about 0.5 Pa at room temperature. The Hall bar structures were patterned into 20 × 10 µm by electron beam lithography (EBL) and Ar ion milling etching technique.

### Characterization

The magnetic properties of the multilayers were measured by vibrating sample magnetometer (VSM, Quantum Design Versalab). $R_H$-H and $R_H$-I loops were measured in physical property measurement system (PPMS, Quantum Design Dyna Cool) or in air by using Keithley 2400, 6221 and 2182 A and the current pulse width time is set to 16 µs. The magnetization switching images were obtained by MOKE microscope under an impulse current without external magnetic field. (Vertisis Technology, MagVision). PNR was measured on the Multipurpose Reflectometer (MR) beamline at the Chinese Spallation Neutron

bar, the NiO/HM interface induces the $\sigma_{sz}$ and the consequent ZFS of magnetization.

It is noted that various antiferromagnet-based ZFS SOT devices have been reported. Table 1 summarizes the comparison between our device and previously reported devices. Since the NiO is an insulator, the mechanism of the $\sigma_{sz}$ is intrinsic without the undesired influence from ambiguous shunting effects in metallic antiferromagnetic layer. Thus, we can expect a higher efficiency in our AFM NiO based SOT derives. Another advantage for this NiO based SOT devices is the

Source (CSNS). The PNR measurements were conducted under an in-plane external field of 1.9 T at room temperature.

## Simulations

All first-principles calculations were performed by density functional theory (DFT) based Vienna ab initio Simulation Package (VASP)[35,36] with the Perdew-Burke-Ernzerhof (PBE) approximation for the exchange-correlation functional and the frozen-core all-electron projector augmented wave (PAW) method for the electron-ion interaction[37]. To include the strong on-site Coulomb interaction of $3d$ orbitals, GGA with a Hubbard $U_{eff} = 5.0$ eV was used for Ni[38,39]. The cutoff energy for the plane wave expansion was set to 500 eV. A Γ-centered $13 \times 13 \times 13$ k-point grid was applied for Brillouin zone sampling. The rhombohedral unit cell of the bulk NiO was used and the atoms were relaxed until the energy and force were converged to $10^{-6}$ eV and 0.001 eV/Å, respectively. The optimized lattice parameter is $a = b = c = 4.19$ Å, consistent with previous studies[40]. Since the (001) surface planes of NiO, MgO and STO are non-polar and charge neutral, the deposition of NiO on these two substrates will not induce significant charge transfer or magnetic reconstruction between NiO film and the substrates. The MgO and STO substrates mainly introduce varied strain effect on the NiO film, in which about 1% tensile strain and 6% compressive strain are imposed on the NiO film, respectively[27,29]. Thus, we only consider the substrate induced strain effect on the magnetic properties of the NiO.

## Data availability

The data that support the findings of this study are available from the corresponding author upon request.

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

## Acknowledgements

This work was partially supported by the National Key Research and Development Program of China (2022YFA1402602), Beijing Natural Science Foundation Key Program (Grant no. Z190007), and National Natural Science Foundation of China (Grant nos. 52061135205, 51971024, 51927802, 51971023, 51971027, 52130103). We acknowledge the computational resources supported by the National Supercomputing Centre (NSCC) Singapore and Centre of Advanced 2D Materials (CA2DM) HPC infrastructure. We are grateful for discussion with Prof. C. Song and he gave us constructive advice.

## Author contributions

M.X.W. and J.Z. contributed equally to this work. X.G.X, M.Y. and Y.J. conceived the project. M.X.W., T.Z.Z., Z.Q.Z., Z.X.G., and Y.B.D. grew materials. M.X.W., T.Z.Z. and Z.Q.Z. fabricated devices. M.X.W., T.Z.Z. and Z.Q.Z. performed magnetic and electrical transport measurements supervised by X.G.X., Y.J. and J.Z. performed the first-principles calculations with M. Y.'s guidance. B.H. and G.Q.Y. performed Moke measurements. J.L.L. and T.Z. performed PNR measurement. A.L. and X.D.H. performed TEM measurement. All authors contributed to discussions. M.X.W. and J.Z. wrote the manuscript with the input from all authors.

## Competing interests

The authors declare no competing interests.
