## [Peer Review File · Nature Communications]

Reviewers' Comments:

Reviewer #1:

Remarks to the Author:

The manuscript deals with the investigation of electric transport in a NiO/Pt/Co/Pt heterostructure where NiO's insulating and antiferromagnetic properties are exploited. The authors interpret their experimental findings in term of a spin orbit torque mechanism which switches the orientation of the Néel vector and thus switches the Hall resistance. The authors also leave the impression that the use of NiO for field-free switching in such heterostructure is new.

My criticism of the manuscript is twofold.

First, the authors need to make a convincing case that the observed hysteresis loops in the Hall-resistance versus current density are indeed the result of reorientation of the Néel vector. This can be doubted for the following reason. Churikova et al. showed in Appl. Phys. Lett. 116, 022410 (2020) that "transverse voltage signals consistent with both the partial switching and toggle switching of the Néel vector in epitaxial Pt/NiO bilayers on Al₂O₃ are also present in Pt/Al₂O₃ in which the AFI is absent." Therefore it seems imperative that the authors make a convincing case that the phenomenon they observe is truly related to switching of the Néel vector. To do so, it requires measurement techniques other than transport which are directly sensitive to the antiferromagnetic order parameter.

Second, the work by Churikova was motivated by claims of Chen et al. Phys. Rev. Lett. 120, 207204 (2018), Moriyama et al., Sci. Rep. 8, 14167 (2018), and Baldrati et al., Phys. Rev. B 98, 024422 (2018) that spin orbit torque switching of the Néel vector happens in NiO/Pt heterostructures. That means, even if the effect observed in the manuscript under consideration here is not an artifact as suggested in Appl. Phys. Lett. 116, 022410 (2020), it is not clear to me how the NiO/Ta/Pt/Co/Pt structure investigated by the authors reveals physics not discussed previously.

Minor:

Line 24: stunting effects -> shunting effects

Line 31: Nèel-> Néel

Reviewer #2:

Remarks to the Author:

The authors demonstrate field-free switching of a Pt-Co-Pt heterostructure by proximity to antiferromagnetic NiO. They attribute the mechanism of the switching to an out-of-plane component of the spin polarisation in Pt that emerges via the interaction with the out-of-plane component of the Néel vector at the interface with NiO. The authors show that the direction of the current-induced out-of-plane exchange depends on the current direction, which excludes thermal effects such as heat-induced strain.

The technological relevance claimed by the authors is not supported by their results since the switching current density is comparable to other field-free switching devices that include antiferromagnetic metals [Nature Nanotechnology 11, 878 (2016)]. Nevertheless, the mechanism of the field-free switching reported here is different from that of these other structures, which has an interest from the scientific point of view.

The authors' model for the current-induced out-of-plane spin polarisation is not convincing.

- A net out-of-plane spin component would only emerge for an uncompensated NiO interface but this is not mentioned and should not be the case for the crystal growth direction considered here.

- The "switching on" of the effect at a sufficiently high current density (fig 4d) as opposed to a gradual increase is not satisfactorily justified and is not in line with the mechanism proposed by the authors.

- The claim that the authors make on the NiO strain affecting the spin polarisation is not well-supported. The authors show that by growing the NiO layer on two different substrates, i.e. MgO and STO, the switching efficiency changes, but I would expect the quality of the growth and especially the NiO-Pt interface to play a determinant role in the process. The NiO crystal quality

and the NiO-Pt interface roughness are not compared for the two samples, so it is hard to make a judgement. It would have perhaps been more convincing to use piezoelectric strain without changing the heterostructure composition.

Table 1 is not useful, it just summarises what has already been stated in the text, a comparison between the switching currents in the different devices would have probably been more useful.

Minor corrections:

Line 24: change stunting to shunting

Line 95: change NiO samples to STO samples.

We would like to thank the reviewers for their careful reviewing of our manuscript and for their constructive comments. We are also thankful for their inspiring comments, which have allowed us to further improve the manuscript.

In the following, we provide a point-to-point response to both reviewers' questions and comments. With this revision, we also attach the revised manuscript and supplementary information addressing the reviewers' comments. We do hope that the reviewers would appreciate our efforts to improve the manuscript following their suggestions and may find our revised manuscript an exciting contribution to the broad scientific community in materials science and spintronic devices.

Please note that our response is highlighted by blue texts and the revisions in the manuscript are highlighted by yellow color.

Reviewer #1:

The manuscript deals with the investigation of electric transport in a NiO/Pt/Co/Pt heterostructure where NiO's insulating and antiferromagnetic properties are exploited. The authors interpret their experimental findings in terms of a spin orbit torque mechanism which switches the orientation of the Néel vector and thus switches the Hall resistance. The authors also leave the impression that the use of NiO for field-free switching in such heterostructure is new.

My criticism of the manuscript is twofold.

First, the authors need to make a convincing case that the observed hysteresis loops in the Hall-resistance versus current density are indeed the result of reorientation of the Néel vector. This can be doubted for the following reason. Churikova et al. showed in Appl. Phys. Lett. 116, 022410 (2020) that "transverse voltage signals consistent with both the partial switching and toggle switching of the Néel vector in epitaxial Pt/NiO bilayers on Al₂O₃ are also present in Pt/Al₂O₃ in which the AFI is absent." Therefore, it seems imperative that the authors make a convincing case that the phenomenon they observe is truly related to switching of the Néel vector. To do so, it requires measurement techniques other than transport which are directly sensitive to the antiferromagnetic order parameter.

Our response:

We thank the reviewer for his/her constructive comments. The reference provided by the reviewer helps us to reconsider the origin of transverse voltage signals. By comparing the device structure in Appl. Phys. Lett. 116, 022410 (2020) with our work, we find that **the origins of the variation of the transverse voltage signals in these two works are different**. In the reference, there is no ferromagnetic layer, and the signal is related with the switching of Néel vector. In contrast, the transverse voltage signals (AHE and R_{H-I} hysteresis loops) in our devices are contributed by the perpendicular magnetic anisotropy of the ferromagnetic Co layer. As shown in Figs. 2(b), 2(d) and Fig. 3(a), we could still observe the AHE and R_{H-I} hysteresis loops in the MgO/Pt/Co/Pt and STO/Pt/Co/Pt samples (without NiO layer). Therefore, in our devices, it is the perpendicular magnetized Co layer that is responsible for the anomalous Hall resistance.

The AFM NiO layer plays important roles in modifying the spin polarization and inducing the magnetization switching of the Co layer at zero-field. In addition, we also observe the magnetization switching process of the Co layer in the MOKE images (Fig 3(e)-(h) in the revised manuscript). Moreover, comparing with the reference, we only applied a pulse current along one direction, in which the variation of resistance caused by electronic migration does not have obvious contribution to the AHE and R_{H-I} loop.

Therefore, the switching in our devices is the magnetization of the Co layer, instead of the Néel vector of NiO. The function of the NiO layer in our devices is to introduce an out-of-plane component of the spin current by the adsorption and reflection of the spin polarization, which drives the zero-field switching of the Co layer, as we discussed in the manuscript.

We also agree with the reviewer that the antiferromagnetic order measurements are necessary besides transport measurements. To clarify the magnetic order of stacking structures, we have characterized the sample of MgO/NiO (5)/Ta (0.2)/Pt (4)/Co (1)/Pt (1) by using polarized neutron reflectivity (PNR) on the Multipurpose Reflectometer (MR) beamline at the Chinese Spallation Neutron Source (CSNS). The PNR

measurements were conducted under an in-plane external field of 1.9 T at room temperature, which thus is only sensitive to in-plane magnetization. As shown in Figure R1(a), the splitting between the spin-up and spin-down neutrons mainly corresponds to the magnetic contribution from Co layer along the in-plane direction. This can be understood by that the out-of-plane Co magnetic ordering can be tilted due to the large external field used in the measurements, which leads to contribution to the measured PNR spectra. Figure R1(b) shows the nuclear scattering length density (nSLD) and magnetization scattering length density (mSLD), which represent the chemical and magnetization profiles of the sample, respectively. It is clear that the magnetization only comes from Co layer and the NiO layer is antiferromagnetic with zero magnetization. According to the PNR results, the thin NiO layer in our samples can be confirmed to be antiferromagnetic.

Figure R1. (a) Polarized neutron reflectivity (R^{++} and R^{--}) of the MgO/NiO (5)/Ta (0.2)/Pt (4)/Co (1)/Pt (1) sample as a function of wave factor (q), where the R^{++} and R^{--} are the reflectivity for polarized neutron parallel or antiparallel to the external field. Open symbols and solid lines represent the experimental data and theoretical fitting, respectively. The inset shows the spin asymmetry, $SA = (R^{++} - R^{--}) / (R^{++} + R^{--})$. (b) Nuclear scattering length density (nSLD) and magnetization profiles (mSLD) of the MgO/NiO (5)/Ta (0.2)/Pt (4)/Co (1)/Pt (1) sample.

Change made:

We have added the above discussions on Page 7 in the revised manuscript and Pages 4-5 in the revised Supporting Information.

We have included Fig. R1 as Fig. S5 in the Supplementary Information.

We have included the mentioned paper (Appl. Phys. Lett. 116, 022410, 2020) as reference (Ref. 23) in the revised manuscript.

Second, the work by Churikova was motivated by claims of Chen et al. Phys. Rev. Lett. 120, 207204 (2018), Moriyama et al., Sci. Rep. 8, 14167 (2018), and Baldrati et al., Phys. Rev. B 98, 024422 (2018) that spin orbit torque switching of the Néel vector happens in NiO/Pt heterostructures. That means, even if the effect observed in the manuscript under consideration here is not an artifact as suggested in Appl. Phys. Lett. 116, 022410 (2020), it is not clear to me how the NiO/Ta/Pt/Co/Pt structure investigated by the authors reveals physics not discussed previously.

Our response:

We agree with the reviewer that the physical mechanism about the Néel vector switching in NiO/Pt system has been reported.

However, the AHE loops and R-I loops are responsible for the magnetization switching of the Co layer but not due to the switching of Néel vector in our system. By designing the MgO/NiO/Ta/Pt/Co/Pt structure, we are aiming to induce a perpendicular polarization component of the spin current by the NiO layer. The AFM NiO layer only absorbs and reflects the spin current generated in the Pt layer, resulting in a perpendicular component polarization of the spin current back to the Pt/Co interface to achieve the magnetization zero-field switching of the Co layer. It means the spin polarization with z direction component is responsible for the zero-field switching. Comparing to the references, **we proposed a new method to achieve the zero-field magnetization switching of the Co layer induced by spin current with z direction component polarization provided by NiO/HM, which has not been reported previously.** In addition, the direction of the Néel vector in the NiO layer can be tuned by the substrate strain, which is demonstrated by the first-principles calculations. **On different substrates, the Néel vector of NiO could be tuned and reoriented due to the lattice strain.**

Change made:

We have included the mentioned paper (Sci. Rep. 8, 14167, 2018) as reference (Ref. 25)

in the revised manuscript.

Minor:

Line 24: stunting effects -> shunting effects

Line 31: Nèel-> Néel

Our response:

We thank the reviewer for pointing out the typos. We have corrected them accordingly and double-checked the writing of the whole manuscript as well.

Reviewer #2:

The authors demonstrate field-free switching of a Pt-Co-Pt heterostructure by proximity to antiferromagnetic NiO. They attribute the mechanism of the switching to an out-of-plane component of the spin polarisation in Pt that emerges via the interaction with the out-of-plane component of the Néel vector at the interface with NiO. The authors show that the direction of the current-induced out-of-plane exchange depends on the current direction, which excludes thermal effects such as heat-induced strain.

The technological relevance claimed by the authors is not supported by their results since the switching current density is comparable to other field-free switching devices that include antiferromagnetic metals [Nature Nanotechnology 11, 878 (2016)]. Nevertheless, the mechanism of the field-free switching reported here is different from that of these other structures, which has an interest from the scientific point of view.

Our response:

We thank the referee for his/her interest on our work and appreciating the novelty of the field free switching mechanism reported in our work.

The authors' model for the current-induced out-of-plane spin polarization is not convincing.

-A net out-of-plane spin component would only emerge for an uncompensated NiO interface but this is not mentioned and should not be the case for the crystal growth

direction considered here.

Our response:

We are thankful for the reviewer's constructive suggestion. We fully agree with the reviewer that we should consider the uncompensated magnetic moments at NiO interface. As the reviewer mentioned, the uncompensated magnetic moments may exist at the interface due to inevitable defects and lattice mismatch. However, our experimental results suggest that the uncompensated magnetic moments can be neglected in our samples.

Firstly, we find that the zero-field switching ratio depends on the lattice strain in our samples, which is consistent with the trend of the strain effect on the Néel vector revealed by the first-principles calculations. Therefore, the out-of-plane spin polarization is dependent on the orientation of the Néel vector after taking surface roughness and sample quality into consideration as shown below in our model. The out-of-plane component of Néel vector could induce the z component of spin polarization via the absorption and reflection process at the interface of NiO/HM. The efficiency will decrease when the Néel vector is tilting to the in-plane direction caused by the lattice strain. If the uncompensated magnetic moments were the main contribution to the zero-field switching, there would not have so large difference between the MgO and STO samples, as the uncompensated magnetic moments are insensitive to the lattice strain.

Secondly, we have studied the magnetic structure of MgO/NiO/Ta/Pt/Co/Pt by polarized neutron reflectivity (PNR). As shown in Figure R2, the NiO layer shows antiferromagnetic ordering. Moreover, there is no obvious net magnetic moments observed at the NiO/Ta/Pt interface (the red line). These results indicate that the uncompensated magnetic moments are negligible in our samples. Thus, we conclude that the uncompensated magnetic moments do not play an important role for the zero-field switching.

Figure R2. (a) Polarized neutron reflectivity (R^{++} and R^{--}) of the MgO/NiO (5)/Ta (0.2)/Pt (4)/Co (1)/Pt (1) sample as a function of wave factor (q), where the R^{++} and R^{--} are the reflectivity for polarized neutron parallel or antiparallel to the external field. Open symbols and solid lines represent the experimental data and theoretical fits, respectively. The inset shows the spin asymmetry, $SA = (R^{++} - R^{--}) / (R^{++} + R^{--})$. (b) Nuclear scattering length density (nSLD) and magnetization profiles (mSLD) of the MgO/NiO (5)/Ta (0.2)/Pt (4)/Co (1)/Pt (1) sample.

It is known that the spin current will be reflected when the spin polarization parallel to the Néel vector and will be absorbed when perpendicular to the Néel vector [Appl. Phys. Lett. 111, 052409 (2017)]. In the NiO/Ta/Pt/Co/Pt heterostructure, the spin current is generated due to the spin Hall effect of Pt, and then reaches the NiO layer. In the MgO based samples, the Néel vector of NiO has an out-of-plane component which can absorb partial spin current and reflect the others. During the absorption and reflection process, the direction of the spin polarization can be affected by the direction of Néel vector, which results in a spin current with out-of-plane component polarization. In contrast, in the STO based samples, the Néel vector of NiO is almost in-plane according to the first-principles calculation results. Therefore, the zero-field switching ratio of the STO sample is much smaller than that of the MgO sample in our manuscript.

Based on the above discussions, we believe that the substrate lattice strain is responsible for the perpendicular component of the Néel vector and consequently the current induced out-of-plane spin polarization. The contribution from the uncompensated magnetic moments is very minor.

Change made:

We have included the above discussions on Page 9 in the revised manuscript.

We have added Fig. R2 and the relevant discussion on Page 4 and 5 in the Supplementary Information.

- The “switching on” of the effect at a sufficiently high current density (fig 4d) as opposed to a gradual increase is not satisfactorily justified and is not in line with the mechanism proposed by the authors.

Our response:

We thank the reviewer for this constructive comment. In the system with PMA magnetic ordering, a spin current with a spin-polarization component in the z direction will generate an anti-damping like torque on the magnetic ordering. When the current is large enough, the anti-damping like torque exceeds the intrinsic damping like torque, thus leading to the threshold phenomenon. The “switching on” results could also be the evidence of the existence of z -component polarized spin current as the similar results reported in *Nat. Mater.* 17, 509–513(2018).

To reveal the effect of the current, we re-measured the R-H loops with different current values and directions from 1 mA to 8 mA. The shift of $|\Delta H(I)|$ with respect to the magnitude of the applied current is shown in Figure R3. $|\Delta H(I)|$ increases with the current after 3 mA, which is consistent with the simulation results reported previously [Nat. Mater. 17, 509–513(2018)]. We have replaced Fig. 4(d) and revised the discussion accordingly in the revised manuscript.

Figure R3. The shift of $|\Delta H(I)|$ with respect of the magnitude of the applied current.

Change made:

We have replaced Fig. 4 with Fig. R3 in the revised manuscript.

- The claim that the authors make on the NiO strain affecting the spin polarisation is not well-supported. The authors show that by growing the NiO layer on two different substrates, i.e. MgO and STO, the switching efficiency changes, but I would expect the quality of the growth and especially the NiO-Pt interface to play a determinant role in the process. The NiO crystal quality and the NiO-Pt interface roughness are not compared for the two samples, so it is hard to make a judgement. It would have perhaps been more convincing to use piezoelectric strain without changing the heterostructure composition.

Our response:

We agree with the reviewer that the quality of the NiO layer and the NiO/Pt interface for the STO samples are important and should be characterized for comparison with the MgO samples. We have studied the surface of NiO and NiO/Pt films by atomic force microscopy (AFM). The AFM images of the as-deposited NiO and NiO/Pt films on MgO and STO substrates are presented in Figure R4. The films are smooth with a R_q value lower than 0.2 nm after NiO layer growth and the roughness are also comparable after the metal layers (Ta (0.2)/Pt (4)) deposition. The detailed data can be seen in the Table R1.

Figure R4. The AFM images of (a) MgO/NiO; (b) MgO/NiO/Ta/Pt; (c) STO/NiO and (d)STO/NiO/Ta/Pt.

Samples	Rq (nm)
MgO/NiO	0.176
MgO/NiO/Ta/Pt	0.175
STO/NiO	0.197
STO/NiO/Ta/Pt	0.180

Table R1. R_q data of the as-deposited NiO and NiO/Pt films on MgO and STO substrates.

Meanwhile, the crystal quality of the NiO layer and the NiO/Pt interface were characterized by HRTEM. The HRTEM images of NiO/Pt interfaces for both MgO and STO samples are presented in Figure R5, which demonstrate the high crystal quality of the NiO layer and the NiO/Pt interface for both samples. We have added the AFM and HRTEM results and the corresponding discussion in the revised manuscript.

Figure R5. The HRTEM images of (a) MgO/NiO; (b) NiO/Pt of MgO sample; (c) STO/NiO and (d) NiO/Ta/Pt of STO sample.

As the reviewer mentioned, many efforts have been paid to tune the magnetization

switching via piezoelectric strain. We agree with the reviewer that the piezoelectric strain is also a possible method to introduce lattice strain to the NiO layer. However, the piezoelectric strain is usually relatively small due to the lattice mismatch (for example PMN-PT, PZT, etc.), which would weaken the piezoelectric strain transition efficiency. More importantly, the electric polarization of the substrate would influence the epitaxial growth of the NiO layer, which would also decrease the strain transition. On the other hand, the piezoelectric strain requires external electric field, which makes the mechanism more complicated. Therefore, in this study, we chose the lattice strain from the substrates MgO and STO to demonstrate the Néel vector induced out-of-plane component of spin polarization. We thank the reviewer's constructive comment and would like to study the piezoelectric strain to modulate the Néel vector in the future.

Change made:

We have included the above discussions on Page 12 in the revised manuscript.

We have added the relevant discussion and Figs. R4, R5 and Table R1 in the revised Supplementary Information (Pages 5-7, Fig. S6 and S7, and Table S1).

Table 1 is not useful, it just summarises what has already been stated in the text, a comparison between the switching currents in the different devices would have probably been more useful.

Our response:

We appreciate the reviewer for the advice. According to the comment, we have added the critical switching current into the Table (as shown in the following Table R1), with which we can compare the critical current density among different mechanisms. It can be seen that the critical switching current in our work is lower than that of the metallic-AFM IrMn-based system and comparable with those of other zero-field switching methods. Therefore, our study provides a mechanism of field-free switching which is different from previous reports.

Items	This work	Previous works
Structure of devices	Antiferromagnetic insulator NiO-based device	Other AFM metallic structures ^[10,11,19-20]
ZERO-FIELD SWITCHING Mechanism	z-direction spin-polarization	a) Exchange bias ^[10,11]
		b) Competing spin current ^[12]
		c) z-direction spin-polarization ^[16-20]
Critical switching current	3~6×10 ¹¹ A/m ²	a) 8~9×10 ¹¹ A/m ² ^[10]
		b) 3×10 ¹¹ A/m ² ^[12]
		c) 6×10 ¹¹ A/m ² ^[20]
Néel temperature	520 K ^[22]	345 K (Mn ₃ GaN) ^[19] , 475 K (Mn ₃ SnN) ^[20]
Shutting effect	No	Yes
Néel vector Regulation	Yes	Yes
z-direction spin-polarization Regulation	Yes (Lattice strain)	No

minor corrections:

Line 24: change *stunting* to *shunting*

Line 95: change *NiO* samples to *STO* samples.

Our response:

We thank the reviewer for pointing out the errors. We have corrected the mistakes accordingly and checked the whole manuscript carefully.

Changing List

1. We added Tanzhao Zhang, Jialiang Li, Tao Zhu, Ang Li and Xiaodong Han to the co-author list.
2. Page 2, Line 5. We changed the word “stuning” to “shunting”.
3. Page 2, Line 12. We changed the word “Nèel” to “Néel”.
4. Page 5, Line 15. We changed the word “NiO” to “STO”.
5. Page 6, Line 22. We added the sentences “Moreover, comparing with previous reports about the Néel vector” according to the reviewer’s comment.
6. Page 9, Line 3. We added the sentences “To clarify the magnetic order of stacking structures, we have characterized the sample of MgO/NiO (5)/Ta (0.2)/Pt (4)/Co (1)/Pt (1) by using polarized neutron reflectivity (PNR).....” according to the reviewer’s comment.
7. Page 12, Line 15. We added the sentences “Furthermore, to compare the quality of NiO layer growth on different substrates ...” according to the reviewer’s comment.
8. We added the References 23, 25, 26 and 27 and renumbered the references accordingly.
9. We added one column to compare the critical switching current in the table I as the reviewer’s suggestion.
10. We added the Note 5, Note 6 and Note 7 in Supporting Information to support our work according to the reviewers’ comments.

Reviewers' Comments:

Reviewer #1:

Remarks to the Author:

I think the authors miss the point which Appl. Phys. Lett. 116, 022410 (2020) makes. The authors state in their reply "In the reference, there is no ferromagnetic layer, and the signal is related with the switching of Néel vector." The point Appl. Phys. Lett. 116, 022410 (2020) makes is that an AHE-like signal can also be present in a Pt/Al₂O₃ system in the absence of an antiferromagnet. This signal, which has nothing to do with any form of magnetism, can also mimic switching of a ferroic order parameter. That is the danger. I feel that a more careful discussion is in order.

I understand now better that the authors report on a device, where the Neel vector orientation is essential for the switching of the ferromagnetic layer, but the Neel vector itself is not reoriented during switching. I suggest to slightly change the title of the manuscript to make very clear which order parameter is switched and which order parameter remains stationary. For example, the authors could say "Stationary perpendicular Néel vector in an antiferromagnetic insulator enables field-free switching of ferromagnetic layer in spin-orbit torque device". In addition it would be useful to find a compact way to express in the title perpendicularity of the Neel vector with respect to what. If the title gets too long by explaining what the reference is to characterize the orientation of the Neel vector, I suggest to leave the word perpendicular out because it adds nothing without explanation. The authors also do not say in the title that the ferromagnet has perpendicular anisotropy so why mentioning that the Neel vector is perpendicular to some reference orientation?

Reviewer #2:

Remarks to the Author:

With respect to the previous version the authors have made a considerable and appreciated effort to provide additional information on the quality of the structures via cross sectional TEM and to confirm the switching of the FM via MOKE measurements.

Although there is a clear contribution from the NiO in the switching of the FM, I still don't believe in the model the authors propose.

First, precession of an in-plane spin polarisation around the Neel vector of NiO will not produce a net spin polarisation in the OP direction because the contribution from the two sublattices will average up. So, unless we have an uncompensated interface, which would not be picked up by PNR measurements, I don't see how this mechanism is possible.

I still don't understand the "switching-on" of the effect at a minimum current density, as shown in Fig.4. The out-of-plane component of the spin polarisation generated via the interaction with NiO is presumably proportional to the current density. This spin polarisation will act as an effective magnetic field on the magnetisation of Co, also proportional to the current density. The observed behaviour is instead reminiscent of a switching event and I find hard to believe that no switching/ domains rearrangement in NiO is involved, and this has an effect on the switching of the FM.

Regardless of the journal in which this will be published, the style with which the manuscript is written is often cumbersome and not grammatically correct and I suggest a language revision.

Response to the reviewers

We would like to thank the reviewers for carefully reviewing our work and for their constructive comments, which have allowed us to further improve the manuscript. In the following, we provide a point-to-point response (highlighted by blue texts) to address all reviewers' questions and comments. With this response, we also attach the revised manuscript and supplementary information (the revisions are highlighted by yellow color).

Reviewer #1 (Remarks to the Author):

R1-1: I think the authors miss the point which Appl. Phys. Lett. 116, 022410 (2020) makes. The authors state in their reply "In the reference, there is no ferromagnetic layer, and the signal is related with the switching of Néel vector." The point Appl. Phys. Lett. 116, 022410 (2020) makes is that an AHE-like signal can also be present in a Pt/Al₂O₃ system in the absence of an antiferromagnet. This signal, which has nothing to do with any form of magnetism, can also mimic switching of a ferroic order parameter. That is the danger. I feel that a more careful discussion is in order.

Our Response:

We thank the reviewer for pointing out our misunderstanding in the previous reply. We agree that the thermal effect could also bring AHE-like or SMR-like signals which have nothing to do with any form of magnetism, according to the previous study (*Appl. Phys. Lett. 116, 022410 (2020)*). To address the reviewer's concerns, we have fabricated two more samples with the stacking structures of NiO (20)/Ta (0.2)/Pt (4) and Ta (0.2)/Pt (4) deposited on MgO(001) substrates, respectively, and measured the relevant R_H-H and R_H-I loops, respectively. As shown in Figure R1, we do not observe any AHE-like or SOT-like loop for these two samples, which is in contrast with the strong loop-like AHE signal shown in Figure 2 and Figure 5 in the manuscript. Therefore, we conclude that the Néel order and domains of NiO have no contribution in our results. The loop-like curves in our work are due to the magnetization switching of the magnetic layer Co.

Figure R1. The R_H - H (a) and R_H - I (b) loops of sample MgO(001)/NiO (20)/Ta (0.2)/Pt (4); The R_H - H (c) and R_H - I (d) loops of sample MgO(001)/Ta (0.2)/Pt (4).

R1-2: I understand now better that the authors report on a device, where the Neel vector orientation is essential for the switching of the ferromagnetic layer, but the Neel vector itself is not reoriented during switching. I suggest to slightly change the title of the manuscript to make very clear which order parameter is switched and which order parameter remains stationary. For example, the authors could say “Stationary perpendicular Néel vector in an antiferromagnetic insulator enables field-free switching of ferromagnetic layer in spin-orbit torque device”. In addition, it would be useful to find a compact way to express in the title perpendicularity of the Neel vector with respect to what. If the title gets too long by explaining what the reference is to characterize the orientation of the Neel vector, I suggest to leave the word perpendicular out because it adds nothing without explanation. The authors also do not say in the title that the ferromagnet has perpendicular anisotropy so why mentioning that the Neel vector is perpendicular to some reference orientation?

Our Response:

We thank the reviewer for his/her constructive suggestion on the title of our manuscript. Following the reviewer's suggestion, we have revised the title as "*Stationary antiferromagnetic insulator interface enables field-free switching of ferromagnetic layer in spin-orbit torque device*" in the revised manuscript.

Reviewer #2 (Remarks to the Author):

With respect to the previous version the authors have made a considerable and appreciated effort to provide additional information on the quality of the structures via cross sectional TEM and to confirm the switching of the FM via MOKE measurements.

Our Response:

We thank the reviewer for his/her appreciation on our effort and new data to improve the manuscript. All his/her comments helped us to further improve the manuscript.

R2-1: Although there is a clear contribution from the NiO in the switching of the FM, I still don't believe in the model the authors propose.

First, precession of an in-plane spin polarisation around the Neel vector of NiO will not produce a net spin polarisation in the OP direction because the contribution from the two sublattices will average up. So, unless we have an uncompensated interface, which would not be picked up by PNR measurements, I don't see how this mechanism is possible.

Our Response:

We appreciate the reviewer's comments on our explanation. After carefully considering the reviewer's comments, we recognized that the explanation on ZFS in NiO based device should be modified. We agree with the reviewer that we should discuss the contribution from the uncompensated magnetic moments as Néel vector would not induce the spin polarization with out-of-plane component due to the average up effect.

Based on the reviewer's comments, we further carried out the PNR measurement

on the STO(001) and MgO(001) samples. As shown in Figure R3, the negative net spin can be observed at the NiO/HM interface on the STO(001) substrate, while it cannot be observed on the MgO(001) substrate. We note that PNR measurements have been used to support the existence of uncompensated spins (E. Zhang, et al. *Phys. Rev. B* 2021, 104, 134408). With these new results and the reviewer's suggestions, we would like to modify our explanation as follows.

The zero-field magnetization switching (ZFS) in our samples can be explained by the internal electric field at the NiO/HM interface and the state of antiferromagnetic interface that influences the spin transport efficiency. As pointed out in the previous study (T. Jin, et al. *ACS Appl. Mater. Inter.* 2022, 14, 9781-9787), the work function difference between the NiO and Pt induces an internal electric field pointing from NiO to Pt. When the spin current arrives at the NiO/HM interface, the internal electric field will interact with the spin, resulting in the spin flip and rotation as well as spin precession. Therefore, the spin polarization will be realigned with out-of-plane component. Finally, the resulting z-polarized spin current will be reflected to the Co layer and drive the ZFS.

As shown in Figure R3 (e), in the MgO(001) sample, the magnetic moments at the interface of NiO/HM are contributed by the NiO layer, in which the spin up and spin down magnetic moments compensate with each other and do not contribute net spins, as demonstrated by the PNR results. Therefore, in the MgO(001) sample, the z-polarized spin current induced by internal electric field will be reflected back at the compensated interface without adsorption. In contrast, in the STO(001) sample, the Néel vector is pulled to the in-plane direction due to lattice strain from the STO substrate. The z-polarized spin current will be absorbed partially at the uncompensated interface, yielding reduced z-polarized spin current acting on the Co layer (shown in Figure R3 (f)). Thus, the changed MAE direction of the antiferromagnetic NiO layer on the STO(001) and MgO(001) and substrates is responsible to the different ZFS ratio. The processes of generation and transportation are depicted in Figure R3 (e) and (f) for the MgO(001) and STO(001) samples, respectively.

Figure R3. Polarized neutron reflectivity (R^{++} and R^{--}) of the (a) MgO(001)/NiO (5)/Ta (0.2)/Pt and (b) STO(001)/NiO (5)/Ta (0.2)/Pt (4)/Co (1)/Pt (1) samples as a function of wave factor (q); Nuclear scattering length density (nSLD) and magnetization profiles (mSLD) of the (c) MgO(001)/NiO (5)/Ta (0.2)/Pt (4)/Co (1)/Pt (1) and (d) STO(001)/NiO (5)/Ta (0.2)/Pt (4)/Co (1)/Pt (1) samples; (e) and (f) are the schematic diagrams of the atoms and the magnetic moments of NiO on MgO(001) and STO(001) substrates, respectively.

To verify this understanding, we have fabricated another sample MgO(111)/NiO (5)/Ta (0.2)/Pt (4)/Co (1)/Pt (1). In this sample, the Néel vector of NiO(111) is along the in-plane direction and the current induced ZFS ratio (about 18%) is comparable with that of the STO(001) case [see Figure R4 (a) and (b)] but much smaller than that on MgO(001) substrate. Similar to the scenario of the STO(001) sample, the spins at the antiferromagnetic interface of MgO(111) sample are uncompensated and the z-polarized spin current has also been absorbed partially. Based on these experimental results, it can be concluded that the ZFS is driven by the z-polarized spin current induced by the internal electric field at the NiO/HM interface. The large ZFS ratio difference between MgO(001) sample and STO(001) sample is caused by the amount

of reflected z-polarized spin current due to different states at the antiferromagnetic interfaces. These discussions have been included in the revised manuscript.

Figure R4. (a) The AHE loop and (b) the SOT-based magnetization switching curve of the Hall bar with the stacking structure of MgO(111)/Ta (0.2)/Pt (4)/Co (1)/Pt (1).

R2-2: I still don't understand the "switching-on" of the effect at a minimum current density, as shown in Fig.4. The out-of-plane component of the spin polarisation generated via the interaction with NiO is presumably proportional to the current density. This spin polarisation will act as an effective magnetic field on the magnetisation of Co, also proportional to the current density. The observed behaviour is instead reminiscent of a switching event and I find hard to believe that no switching/domains rearrangement in NiO is involved, and this has an effect on the switching of the FM.

Our Response:

We appreciate the reviewer's comments. In our work, a critical current is needed to overcome the intrinsic damping effect of Co layer inducing the "switching-on" phenomenon. Therefore, the "switching-on" (as marked by the purple arrow in Figure R5) effect is just used to prove the existence of z-polarized spin current in our work. Similar "switching-on" effect at a minimum current density has been reported in the previous studies. For example, in the study (S. C. Baek, et al. *Nat. Mater.* 2018, 6, 17), it has been shown that "The current with a particular polarity generates a spin current

flowing out-of-plane with a spin z component and generates an anti-damping torque for the perpendicular magnetization. Anti-damping torque causes an abrupt increase in the loop shift as a function of $I_{d.c.}$ at a threshold above which it exceeds the intrinsic damping, as in conventional spin-transfer torque studies (also indicated by down arrows in our modelling results shown in Figure R5 as below)”. We note that this effect has been used in the various studies to show the existence of the z -polarized spin current (S. Hu, et al. *Nat. Commun.* 2022, 13, 4447; I. H. Kao, et al. *Nat. Mater.* 2022, 21, 1029–1034; T. Jin, et al. *ACS Appl. Mater. Inter.* 2022, 14, 9781-9787).

Figure R5. Micromagnetic simulation results of the loop-shift field ΔB_S versus d.c. current density for cases of the interface-generated spin current ($\sigma = y + \delta z$, where δ is the ratio of the spin z component to the spin y component; square symbols) and of the bulk spin Hall effect ($\sigma = y$; open circular symbols). The horizontal axis is normalized by J_{c0} , which is the threshold switching current density for spin currents with only spin z component. The loop-shift field ΔB_S is defined as the difference in the centres of the hysteresis loop for an in-plane d.c. current $+I_{d.c.}$ and $-I_{d.c.}$. We note that B_x is zero (20 mT) for the case of the interface-generated spin current (bulk spin Hall effect). The arrow of “switching on” is used to mark the threshold current value to overcome the intrinsic damping. Adapted from (S. C. Baek, et al. *Nat. Mater.* 2018, 6, 17)

On the other hand, we would like to clarify the contribution of Néel order and domains of NiO to the magnetic switching in our work. The theoretical critical current threshold for NiO Néel order switching is up to $5.8 \times 10^{12} \text{Am}^{-2}$ (*Phys. Rev. Lett.* 2018,

120, 207204), which is almost one order larger than our work (a maximum current of $6 \times 10^{11} \text{A/m}^2$). In experiment, Néel order switching has been realized using a lower threshold current, which however is accompanied with a pulse width as long as 1 ms (*Phys. Rev. Lett.* 2018, 120, 207204 and *Phys. Rev. Lett.* 2019, 123, 177201). In our study, a low current pulse with short width (16 μs) was used, leading to a much-reduced possibility of the Néel vector switching in our measurement. Thus, the Néel vector in this study is not expected to reorient during the process of the Co magnetization switching.

In order to further confirm the contribution of NiO Néel order, we have prepared and measured a control sample with the stacking structure of MgO(001)/NiO(20)/Ta(0.2)/Pt(4). As shown in Figure R6, we do not observe the AHE loop or SOT loop for the control sample, which indicates the Néel order and domains of NiO have no contribution in our results. The loop-like curves in our work are due to the magnetization switching of the magnetic layer Co.

Figure R6. The R_H -H (a) and R_H -I (b) loops of sample MgO(001)/NiO(20)/Ta(0.2)/Pt(4)

R2-3: Regardless of the journal in which this will be published, the style with which the manuscript is written is often cumbersome and not grammatically correct and I suggest

a language revision.

Our Response:

We have revised the language throughout the manuscript under the help of a colleague who is good at English writing.

List of Changes

1. We changed the title of the manuscript as “Stationary antiferromagnetic insulator interface enables field-free switching of ferromagnetic layer in spin-orbit torque device”.
2. We added the Note 3 in the supplementary to response the reviewer’s comments
3. We moved the Note 5 to the manuscript and added the Fig. 6 to illustrate our model.
4. Page 7, Line 8. We added the sentence “*The loop-like signal also cannot be observed in the samples without Co layers (See Supplementary Fig. 3). Therefore, it can be concluded that the loop-like curves are only a response to the magnetization switching of the magnetic Co layer. On the other hand, the predicted critical current threshold for NiO Néel order switching is up to $5.8 \times 10^{12} \text{ Am}^{-2}$ [26], which is almost one order of magnitude larger than in this work (a maximum current of $6 \times 10^{11} \text{ Am}^{-2}$). In experiment, Néel order switching has been realized using a lower threshold current, which however is accompanied with a pulse width as long as 1 ms [24,26]. In our study, a low current pulse with short width (16 μs) was used, leading to a much reduced possibility of the Néel vector switching in our measurement.*” according to the reviewer’s comment.
5. Page 8, Line 10. We rearranged this paragraph to highlight the key points.
6. Page 10, Line 19. We added the sentence “*While in the PNR measurement of STO(001)/NiO (5)/Ta (0.2)/Pt (4)/Co (1)/Pt (1) sample, as presented in Figs. 6(b) and (d), the negative net spins can be obtained at the NiO/HM interface on STO(001)/NiO(5) substrate. It indicates that the interface of NiO/HM is uncompensated in the STO(001)/NiO (5) samples.*” to explain the new experiment results.
7. Page 11, Line 3. We added the sentence “*Due to the work function difference*

between the NiO and Pt, there is an internal electric field pointing from NiO to Pt.” to modify our model.

8. Page 11, Line 6. We added the sentences “*The initial polarization direction of the spin current is along the in-plane direction and perpendicular to the current. When the spin current arrives at the NiO/HM interface, the internal electric field will interact with the spin, resulting in the spin flip and rotation as well as spin precession. Therefore, the spin polarization will be realigned with out-of-plane component. Then, the z-polarized spin current will be reflected to the Co layer and drive the ZFS. In the MgO(001) sample, the process of generation and transportation of z-polarized spin current are illustrated in Fig. 6(e). In this case, the interface of NiO/HM contains both spin up and spin down Ni moments, so the magnetic moments compensate each other and could not contribute net spins. Therefore, in the MgO(001) sample, the z-polarized spin current induced by an internal electric field will be reflected back at the compensated interface without absorption.*” to modify the original model.
9. Page 12, Line 1. We modified this paragraph according to the modified model.
10. Page 13, Line 5. We added the paragraph “*Based on the above analysis, it can be concluded that the ZFS in our samples is related to the internal electric field and interfacial states of NiO/HM. To further verify this model, we also fabricated another sample with the stacking structure of MgO(111)/NiO (5)/Ta (0.2)/Pt (4)/Co (1)/Pt (1). In this sample, the crystal orientation of epitaxially grown NiO is [111]. Therefore, the Néel vector of NiO is along the in-plane direction since the easy plane is along the (111). In this case, the SOT induced ZFS ratio (see in Supplementary Figs. 8) is comparable with that of the STO(001) sample. The switching ratio of ZFS is calculated to be about 18% on the MgO(111) substrates. Considering that the Néel vector in MgO(111)/NiO (5) and STO(001)/NiO (5) samples is along the in-plane direction, the uncompensated interface will absorb the z-polarized spin current partially, similar to the STO(001) case.*” to explain the new experiment results.
11. We have revised the language throughout the manuscript under the help of a

colleague who is good at English writing and marked in the manuscript.

12. We added the “Data Availability” section and revised the manuscript according to the formatting instructions.

Reviewers' Comments:

Reviewer #2:

Remarks to the Author:

I appreciate the additional efforts made by the authors in understanding the mechanism. The experimental results are intriguing enough to justify a publication.

However, the English in which the paper is written is sometime cumbersome, particularly the explanation of the mechanism the authors provide is very hard to follow.

I don't think the title is appropriate, it is obvious that the interface is stationary. The present title is a modified version of the first reviewer's suggestion, but as it is, it doesn't make much sense.

So, I would suggest a careful revision of the English and a different title, for example "field-free SOT switching of a perpendicular ferromagnet via an out-of-plane spin polarisation induced at the Pt-NiO interface."

Response to the reviewers

We would like to thank the reviewers for carefully reviewing our work and for their constructive comments. In the following, we have provided a point-by-point response (highlighted in blue texts) to address all reviewers' questions and comments. Along with this response, we have also attached the revised manuscript and supplementary information (the revisions are highlighted by yellow color).

Reviewer #2 (Remarks to the Author):

R2-1: I appreciate the additional efforts made by the authors in understanding the mechanism. The experimental results are intriguing enough to justify a publication. However, the English in which the paper is written is sometime cumbersome, particularly the explanation of the mechanism the authors provide is very hard to follow.

Our Response:

Thanks for the reviewer's suggestion, we have polished the writing and simplified the explanation of mechanism in the revised manuscript.

R2-2: I don't think the title is appropriate, it is obvious that the interface is stationary. The present title is a modified version of the first reviewer's suggestion, but as it is, it doesn't make much sense. So, I would suggest a careful revision of the English and a different title, for example "field-free SOT switching of a perpendicular ferromagnet via an out-of-plane spin polarisation induced at the Pt-NiO interface."

Our Response:

Based on the reviewer's suggestion, we have revised the title as "*Field-free spin-orbit torque switching via out-of-plane spin-polarization induced by an antiferromagnetic insulator/heavy metal interface*" in the revised manuscript.

List of Changes

1. We changed the title of the manuscript as "*Field-free spin-orbit torque switching via out-of-plane spin-polarization induced by an antiferromagnetic insulator/heavy metal interface*".
2. We have revised the writing of the manuscript.